# Effects of Vegetation Restoration on Soil Nitrogen Fractions and Enzyme Activities in Arable Land on Purple Soil Slopes

**DOI:** 10.3390/plants12244188

**Published:** 2023-12-18

**Authors:** Bowen Li, Yi Zhang, Yuxin Yao, Peng Dang, Taimoor Hassan Farooq, Xiaohong Wu, Jun Wang, Wende Yan

**Affiliations:** 1Technology in Forestry and Ecology in South China, Central South University of Forestry and Technology, Changsha 410004, China; 19174164183@163.com (B.L.); emailofzhangyi@163.com (Y.Z.); yyx555999@163.com (Y.Y.); dangpeng@csuft.edu.cn (P.D.); wxh16403@163.com (X.W.); 2Bangor College China, A Joint Unit of Bangor University and Central South University of Forestry and Technology, Changsha 410004, China; t.farooq@bangor.ac.uk; 3Lutou National Station for Scientific Observation and Research of Forest Ecosystem in Hunan Province, Yueyang 414000, China

**Keywords:** vegetation restoration, effective nitrogen components, N-cycle enzymes, purple soil, *Camellia oleifera*

## Abstract

Purple soils are greatly representative of ecologically fragile soils in southern China, yet the impact of vegetation restoration processes on the nitrogen (N) availability in purple soil ecosystems is still unclear. In this study, the soil nutrient content, available N fractions (including microbial biomass N (MBN), ammonium N (NH_4_^+^-N), nitrate N (NO_3_^−^-N), and total dissolved N (TDN)), and enzyme activities (including urease (URE), nitrate reductase (NR), and nitrite reductase (NIR)) involved in N mineralization and immobilization were investigated across the three vegetation-restoration measures: *Camellia oleifera* monoculture, *Camellia oleifera* ryegrass intercropping, and *Camellia oleifera* intercropping with weeds. The results showed that the *Camellia oleifera* monoculture mode considerably enhanced the accumulation and availability of soil N and modified the proportion of available N fractions in arable land situated on purple soil slopes, compared to the intercropping mode, the physical, chemical, and microbiological properties of soil demonstrated more pronounced effects due to the *Camellia oleifera* monoculture vegetation-restoration measures. However, soil nutrient loss is faster on set-aside land and in crop monocultures, and intercropping restoration measures are more beneficial for soil and water conservation under timely fertilization conditions. The soil URE, NR, and NIR activities and MBN content in the *Camellia oleifera* monoculture model were significantly higher than in the control check sample. Soil N transformation occurs through the combined influence of chemical and biological processes. The relationships between the activities of the three soil enzymes studied and the contents of various components of soil nutrients and effective N displayed significant differences. Notably, URE had a highly significant positive correlation with TOC. There is a strong positive correlation between NR and TN, NIR and TDN, NO_3_^−^-N, and NH_4_^+^-N. Our findings suggest that vegetation restoration improved the soil N availability and its enzyme activities in purple soils, making an essential contribution to the restoration and sustainability of purple soil ecosystem functions.

## 1. Introduction

Vegetation restoration has been recognized as an effective strategy for preventing soil erosion and improving soil fertility levels by regenerating degraded ecosystems and increasing net ecosystem productivity through changes in the composition and cover of the vegetation community [1]. Previous studies have confirmed that vegetation restoration can modify soil nutrient sequestration by changing the microclimate and physical structure of the soil, thus, ultimately, affecting it [2]. 

Nitrogen (N) is the main limiting element in soils, influencing plant growth and maintaining ecosystem stability, and it plays a key role in soil fertility and land productivity [3,4]. In terrestrial ecosystems, plant-available N is mainly derived from soil N (e.g., the mineralization of soil N); therefore, soil N availability is an important parameter in determining how an ecosystem is functioning. Soil N content and its different components can be influenced by management practices [5], land-use changes [6], vegetation types, etc. [7,8]. In the process of vegetation restoration, plants provide C and N to the soil mainly through root secretions and plant residues, influencing soil N inputs and significantly altering soil properties [9]. At the same time, vegetation restoration effectively prevents N loss through soil erosion, which is conducive to the accumulation of N pools in the soil. Weintraub et al. [10] found that the microbial biomass N and nitrate N contents of grassland soils were lower than those of forests, and Wang et al. [11] found that the total N and microbial biomass N contents of soils in pure fir forests were lower than those in mixed fir–alder and mixed fir–red pine forests. However, some studies also found that there were no significant in differences in the soil total and ammonium N contents between different vegetation types [7,8]. Thus, the study of soil N content in complex agroforestry systems accroding to different vegetation types is inconclusive. Furthermore, soil N transformation is a complex process involving biotic (e.g., soil enzymes) and abiotic (soil nutrients, pH, etc.) factors, which require the involvement of specific soil enzymes at each stage of N transformation. The level of N-cycle-related enzyme activities can characterize the capacity of N supply and transformation in the soil and, to some extent, reflect the status of N uptake and utilization by plants [12,13]. Vegetation restoration has also been found to increase N cycle enzyme activity and reverse soil microbial N limitation in subtropical forest soils [14,15], suggesting that the effects of vegetation restoration can increase soil N turnover and improve the ability to supply N. Yan et al. [16] reported a significant increase in N-cyclase in ecosystem soils during the early stages of restoration, which remained stable as the restoration process progressed. These different results may be due to differences in vegetation cover types, resulting in changes in plant residues, root systems and secretions, and soil properties [17,18,19]. Therefore, the effect of vegetation restoration on soil N-cycle enzyme activities in a given environment remains unclear.

Located in the south-central part of Hunan Province and the middle reaches of the Xiangjiang River, the Hengyang purple soil hills and slopes cover an area of 1625 × 105 hm^2^ and represent one of the most severe ecological environments in Hunan Province and one of the representative ecological disaster-prone areas in southern China. The region has severe soil erosion, sparse vegetation, exposed bedrock, and in some areas almost no soil development layer; the ecological environment is very poor, and it is very difficult to restore vegetation, and the restoration and reconstruction of vegetation in this region is a long-term and labor-intensive project [20,21]. Since the 1990s, governments at all levels have been restoring and re-establishing vegetation in the degraded ecosystems of the purple soil hillsides in Hengyang, and because the natural succession of vegetation takes a long time, generally up to several decades or even longer [22,23], appropriate artificial regulation will accelerate the process of vegetation recovery and significantly shorten the recovery time [24]. Therefore, vegetation restoration should not passively wait for the natural recovery of vegetation but should reasonably select plants that are compatible with the local ecological environment and artificially configure the plant community based on the ecological compatibility of each plant to influence the community succession process.

This study investigates the effects of soil N fractions and N-cycle-related enzyme activities on sloping cultivated land with purple soil under three vegetation restoration modes and control checks. We hypothesized that (1) vegetation restoration patterns enhance soil N accumulation and availability and its enzyme activities; and (2) N-related enzyme activities may have a significant positive correlation with the available N fraction contents. The purpose of this study is to provide a theoretical basis and practice for soil and water conservation in purple soil areas and the precise restoration of degraded land in purple soil.

## 2. Results

### 2.1. Changes in Soil N Fractions in Different Vegetation Types

The soil TDN, NO_3_^−^-N, NH_4_^+^-N, and MBN were significantly affected by vegetation restoration (Figure 1A–D). The soil TDN content was significantly higher in CO (*Camellia oleifera* monoculture) and CK (the control check) than in CR (*Camellia oleifera* ryegrass intercropping) and CW (*Camellia oleifera* intercropping with weeds), with a general trend of CO > CK > CR > CW (Figure 1A). The NH_4_^+^-N content in the three vegetation restoration treatments showed a trend of CO > CR > CW (Figure 1B), and its content in CO was significantly higher than CK in April and July (*p* < 0.05, Figure 1B). Compared to CK soil, the two vegetation restoration modes of CR and CW had significantly lower NO_3_^−^-N contents in April and July, while its content in CO was significantly higher than CK in April (*p* < 0.05, Figure 1C). Specifically, the soil MBN content in the three vegetation restoration modes showed significantly higher than CK in April and July, ranking CO > CR > CW > CK (*p* < 0.05, Figure 1D).

### 2.2. Changes in Soil N Cycle Enzyme Activities in Different Vegetation Types

Significant differences in soil urease (URE), nitrate reductase (NR), and nitrite reductase (NIR) activities were found among the three vegetation-restoration measures and CK plots (*p* < 0.05, Figure 2). The soil URE activity in CR and CW was significantly higher than CK in April, and its activity in CO was significantly higher than CK in July (*p* < 0.05, Figure 2A). The soil NR activity in three vegetation restoration modes was significantly higher than CK, ranking CW > CO > CR = CK in April and CO > CW > CR > CK in July (Figure 2B). Compared to CK, the NIR activities of the three vegetation restoration treatments were not significantly different in April (*p* > 0.05), whereas in July the NIR activities of CO and CW were significantly greater than those of CK, with the NIR activities of CO and CW being 38.2% and 47.1% greater than those of CK, respectively (*p* < 0.05, Figure 2C).

### 2.3. Correlation Analysis

Pearson’s correlation analysis showed a significant correlation between soil nutrients and enzyme activity. There was a significant negative correlation between soil URE activity and NO_3_^−^N content with a correlation coefficient of −0.390 (Figure 3). There was a highly significant positive correlation between soil NR and TN content, with a correlation coefficient of 0.464, and a highly significant negative correlation with NO_3_^−^N content, with a correlation coefficient of −0.449 (Figure 3). There was a significant positive correlation between the soil NIR and TDN, NO_3_^−^-N, and NH_4_^+^-N contents, with correlation coefficients of 0.361, 0.354, and 0.409, respectively (Figure 3).

## 3. Materials and Methods

### 3.1. Experimental Site

The study site is located in Changning City, Hunan Province, China (26°28′ N, 112°21′ E) (Figure 4). The dominant vegetation in the study area is *Camellia oleifera*, known as tea oil. The study site is situated at an average altitude of 170 m above sea level. It has a subtropical monsoon climate with average annual sunshine of 1577.6 h, an average annual temperature of 16–24 °C, an annual frost-free period of 295 days, and a rainy season mainly from April to June. The average annual rainfall over the last three years has been 610–1318 mm, with 445–912 mm from April to September during the growing season. The area is dominated by soils developed on purple sandstone.

### 3.2. Materials and Experimental Design

In mid-May 2022, nine runoff plots were established for the three selected vegetation-restoration measures under similar conditions of elevation, slope, slope position, soil texture, etc. The outflow plots were constructed with corrosion-resistant plastic partitions, and the sample area of each outflow plot was 5 m × 15 m (75 m^2^), with a plot spacing of 0.5 m. The vegetation-restoration measures were *Camellia oleifera* monoculture (CO), *Camellia oleifera* ryegrass intercropping (CR), and *Camellia oleifera* intercropping with weeds (CW), and three plots were replicated for each vegetation restoration type. Three outflow plots were selected as control plots (CK) of bare ground with the same background conditions as the sampling site and no vegetation-restoration measures. Perennial ryegrass is in local demand as fodder and also has the advantage of being a high-quality forage that can be harvested as silage. In addition, ryegrass is tufted, with a well-developed root system, and its fibrous roots are mainly distributed in the top 15 cm of soil, which has a good soil and water conservation effect and can effectively intercept nutrient loss from the soil. Natural grasses, like artificial grasses, have the effect of improving soil physical properties, preventing soil erosion, and reducing environmental pollution, and have become one of the most widely used soil management methods in many countries and regions of the world. The dominant vegetation at the study site is tea oil, which is a taproot plant with a main root that can penetrate to a depth of 2–3 m, so there is no competition between the two root systems, and the tea oil yield is not affected. Among the naturally occurring weeds in the intercropping of tea oil and weeds (CW), there are mainly six plants: *lemongrass* (*Gramineae*, *Citronella* spp.), *chili* (*Camphoraceae*, *Mugilaceae*), *manzanita* (*Ribes family*, *Manzanita*), *abaca* (*Euphorbiaceae*, *Hypocarpus*), *sarsaparilla* (*Sarsaparillaceae*, *Sarsaparilla*), and *saltbush* (*Lacertidae, Saltbush*).

### 3.3. Soil Sampling and Analysis

Ryegrass was planted in December 2022, and soil samples were collected on two separate occasions in April 2023 (ryegrass flourished) and July 2023 (ryegrass wilted). We randomly selected a sampling point in each runoff sub-district slope (upper, middle, and lower slopes) in each slope, and removed the surface layer of apoptosis, with a diameter of 3 cm soil auger, to collect 0–10 cm thickness soil samples, and for each sampling point, we took three replicates of soil samples and mixed them into one sample in a plastic bag. The collected soil samples were sieved on-site through a 2 mm sieve and stored in two parts in sealed bags: one part was stored in a refrigerator at 4 °C for the determination of MBN and soil enzyme activities, and the other part was naturally dried indoors and sieved through a 0.149 mm sieve for the determination of basic soil chemical properties. Soil TOC was quantified via oxidation using K_2_Cr_2_O_7_-H_2_SO_4_ followed by titration with FeSO_4_ [25]. Soil TN was determined using the semi-micro Kjeldahl method, and the flow injector and TP were determined via NaOH melt, molybdenum antimony anticolor development, and UV spectrophotometry [26]. The three indexes measured above are shown in Table 1. TDN was determined via UV spectrophotometry with alkaline potassium persulphate digestion; NO_3_^−^-N was determined using ion chromatography (ICS-1100); NH_4_^+^-N was determined via the indophenol blue colorimetric method; MBN was first fumigated with CHCL_3_ and then extracted with K_2_SO_4_. The procedure for assessing the chemical properties of soils followed the guidelines outlined in Bao’s *Agrochemical Analysis of Soils, Third Edition* [27]. The activity of each soil enzyme was analyzed according to the instructions of the soil enzyme assay kit (Beijing Solabao Biotechnology Co., Ltd., Beijing, China) [28]. The soil pH was determined using the potentiometric method with a water/water mass ratio of 2.5:1 [29].

### 3.4. Statistical Analysis

Excel 2019 was used for the calculation and preliminary analysis of the data. One-way analysis of variance (ANOVA) was performed on the data using IBM SPSS Statistics 26.0 software to analyze the significance of the differences in the indicators between the different treatment measures. Bivariate correlation analysis was used to calculate Pearson’s correlation coefficients between the two indicators. Plotting was conducted with GraphPad Prism 8 software.

## 4. Discussion

### 4.1. Effect of Vegetation-Restoration Measures on the Content of Soil N Fractions

Different soil textures and chemical properties of organic carbon sources and a range of microbial activities affect the content and distribution of N fractions in soils to varying degrees [30]. The restoration of vegetation will certainly improve some soil microenvironmental elements, such as temperature and humidity, water exchange, and the decomposition of dead wood, and the secretion of organic acids through root secretions will also accelerate the conversion of soil insoluble substances into soluble substances, thus improving the conversion capacity of soil N [31,32]. The results of this study show that the response patterns of soil N components to vegetation-restoration measures were significantly different. In both periods, the total soluble N content of the soil from the *Camellia oleifera* monoculture restoration and the control sample plots did not differ significantly, and both were significantly higher than that of the *Camellia oleifera* ryegrass intercrop and the *Camellia oleifera* weed intercrop, which may be due to the excessive growth of ryegrass and weeds in April, which absorbed and fixed a large amount of N. The total soluble N content in the soil was therefore lower, while the wilted ryegrass and weeds in July had not yet returned the nutrients to the soil in time, so the total soluble N content in the soil was even lower.

Soil mineral N contains two types of N, ammonium N and nitrate N [33], and some studies have found a positive correlation between soil ammonium N and nitrate N [34,35]. The results of this study also showed the same trend for these two mineral N nutrients in the three vegetation-restoration treatments and the control sample plots in April, with the *Camellia oleifera* monoculture treatment being significantly higher than the control sample plots and the other two plant treatments. This is because weeds and ryegrass may compete with *Camellia oleifera* for N resources and have faster growth rates. Weeds and ryegrass may have a greater capacity to absorb and utilize N in the soil, resulting in lower levels of both N minerals in the soil. Tea tree is an N-fixing plant, and its roots have symbiotic rhizobacteria that can convert N in the air into organic N in the soil, which helps to increase the N contents of the soil so that the ammonium N and nitrate N content of the soil in the *Camellia oleifera* monoculture are always higher than those of the control site. In contrast, two opposite scenarios were observed in July: (1) the ammonium N was significantly lower in the control plots than in the restoration plots in the three treatments, and (2) the nitrate N was significantly higher in the control plots than in the restoration plots in the three treatments. This is because, during the wilting period, there is no vegetation cover at the control site, and the soil is exposed to the environment and susceptible to oxidation. In addition, there is no plant uptake or use of N from the soil, leaving the soil with relatively high levels of nitrate N. The N demand of oilseed rape, ryegrass, and weeds also reduces the nitrate N content of the soil. Plants that extract nutrients from the soil for their growth have been shown to preferentially use the nitrate N in the soil, thereby reducing the net nitrate N remaining in the soil [36,37,38,39]. The low TDN, NO_3_^−^-N, and NH_4_^+^-N contents of the two plant-restoration treatments, artificial grass planting, and natural grass planting, as shown in the results of this study, indicate that herbaceous plants, which have a significantly lower root biomass than woody plants, have a significantly weaker ability to improve soil N accumulation. This result may also be related to the fact that ryegrass is a cash crop and anthropogenic mowing and harvesting significantly reduce apomictic decomposition and soil N return, resulting in lower N levels for the two treatments [40]. Hengyang purple soil hilly slopes have severe soil erosion and soil nutrients are easily lost through runoff [20]. Li Tao et al. [41] pointed out that the green manure intercropping method can reduce soil nutrient loss more than plant monocropping as well as land abandonment, and the intercropping method seems to be more advantageous in terms of soil and water conservation in purple soil sloped arable land under the condition of reasonable fertilizer application.

Soil microorganisms are living components of the soil, are extremely sensitive to various changes in the soil environment, and can fully reflect the ecological function of the biome. The microbial population is not only an important source of soil nutrients but also an important carrier of soil nutrient fixation [42]. The results of this study showed that the soil microbiome N content of the three planted vegetation-restoration treatments was significantly higher than that of the control sample site in both periods, which was due to the lack of vegetation cover and low input of plant residues and organic matter at the control sample site, resulting in relatively low soil organic matter content. The planting of oilseed rape and ryegrass or the planting of oilseed rape and weeds may have increased the soil organic matter content through the deposition of fallen leaves, dead plant material, and root secretions. Organic matter is an important source of nutrients for microbial growth and activity, so areas with higher soil organic matter tend to have richer microbial communities. Studies by Liang Yueming [43] and Zheng Hua et al. [44] showed that the soil microbiota increased with vegetation restoration, and the present study is consistent with the results of previous studies.

### 4.2. Effects of Vegetation-Restoration Measures on the Activities of Soil N Cycle Enzyme Activities

Soil enzyme activity refers to the ability of soil enzymes to catalyze soil development and fertility formation, and its activity can indicate the strength of soil microbial activity, which can be used as an important biological indicator of soil quality and health [45,46]. Soil enzyme activities such as nitrate reductase, nitrite reductase, and urease are closely related to the intensity of soil N transformation and soil N supply capacity [47,48]. In this study, we found that vegetation restoration significantly increased soil urease, nitrite reductase, and nitrate reductase activities. This is consistent with previous studies that have confirmed that soil enzyme activities are affected by vegetation restoration [27,49].

In this study, the activities of the above three soil enzymes were significantly enhanced through the three vegetation-restoration treatments to varying degrees, indicating that vegetation-restoration treatments can significantly improve the efficiency of soil nutrient use by enhancing the activities of nitrate reductase, nitrite reductase, and urease, which also provide a better environment for microbial growth and activities. Soil urease acts on the carbon and N bonds of soil organic matter to hydrolyze urea to ammonia. Yang Changming et al. [50] showed that soil urease activity was significantly correlated with TOC content under different land-cover patterns. The three vegetation-restoration treatments in this study significantly increased soil urease activity in both periods, suggesting that vegetation-restoration treatments significantly affected soil TOC content by increasing soil urease activity. Nitrate reductase and nitrite reductase are important enzymes involved in soil N denitrification [51]. Nitrate reductase is a specific enzyme that catalyzes the reduction of nitrate to nitrite under aerobic conditions, denitrification produces the greenhouse gas N_2_O under anaerobic conditions, and nitrite reductase catalyzes the conversion of NO_2_^−^ to NO or NH_3_ in soil [52]. The three vegetation-restoration treatments in this study significantly increased soil nitrate reductase activity in both periods, whereas soil nitrite reductase did not change significantly in either period. This may be because nitrate reductase activity can be higher when the nitrate availability is higher, whereas nitrite reductase activity is lower. This is because nitrate reductase provides more substrate for nitrite reductase to catalyze further reduction by reducing nitrate to nitrite.

### 4.3. Correlation of Soil N Cycle Enzyme Activities with Soil Environmental Factors

Changes in aboveground plant species and soil chemical properties are the main causes of changes in soil enzyme activities [53]. Luo Mingxia et al. [54] showed that soil pH, nutrient efficiency, and soil microbial content have a certain correlation with soil enzyme activity. Urease is an important soil hydrolytic enzyme that plays a key role in the breakdown of urea and the N cycle in soil [55,56]. The results of this study show that urease activity was significantly positively correlated with soil TOC content but not significantly correlated with soil N content. This may be related to the fact that the soils of the purple sloped cultivated area are poor, with low organic matter, few substrates for soil urease decomposition, and fewer microbial species related to soil urease production [57,58]. The highly significant positive correlation between soil nitrate reductase and TN content and the highly significant negative correlation with NO_3_^−^N content indicated that nitrate reductase activity was directly related to the characteristics of the soil N cycle. When the total N content of the soil was high, it indicated that the N supply in the soil was relatively sufficient and the N cycle was active. Nitrate reductase regulates the conversion of nitrate N in the soil, and its activity may be regulated by the negative feedback of nitrate N concentration to maintain the balance of the soil N cycle. There was a significant positive correlation between TDN, NO_3_^−^-N, and NH_4_^+^-N contents and soil nitrite reductase, further confirming that the level of nitrite reductase activity is closely related to the efficiency of N utilization and that some microorganisms can directly utilize these forms of N to meet their N requirements when the soil contains high levels of nitrate and ammonia N. Nitrite reductase activity may subsequently increase to accommodate changes in the use patterns of different forms of N.

## 5. Conclusions

Vegetation restoration significantly improved N distribution and accumulation in arable land on purple slopes, significantly affected soil N availability, and altered the contents of soil active N fractions (TDN, NO_3_^−^-N, NH_4_^+^-N, and MBN). The effects of different vegetation-restoration measures on soil active organic N fractions were significantly different. The effects of vegetation-restoration measures on the physical, chemical, and microbiological properties of the soil were more pronounced for the *Camellia oleifera* monoculture than for the intercropping pattern. However, soil nutrient loss is faster on set-aside land and crop monocultures, and intercropping restoration measures are more beneficial for soil and water conservation under timely fertilization conditions. In addition, vegetation restoration increased the activities of nitrate reductase, urease, and nitrite reductase and increased the N content of soil microbial biomass. Soil N transformations occurred under the combined influence of chemical and biological processes. The correlations between the three soil enzyme activities studied and the contents of soil nutrients and effective N components were significantly different, with highly significant positive correlations between urease and TOC, highly significant positive correlations between nitrate reductase and TN, and highly significant positive correlations between nitrite reductase and TDN, NO_3_^−^-N, and NH_4_^+^-N. This suggests that vegetation restoration of cultivated land on purple soil slopes can affect soil N availability and supply by altering soil nutrient contents and effective N components.

## Figures and Tables

**Figure 1 plants-12-04188-f001:**
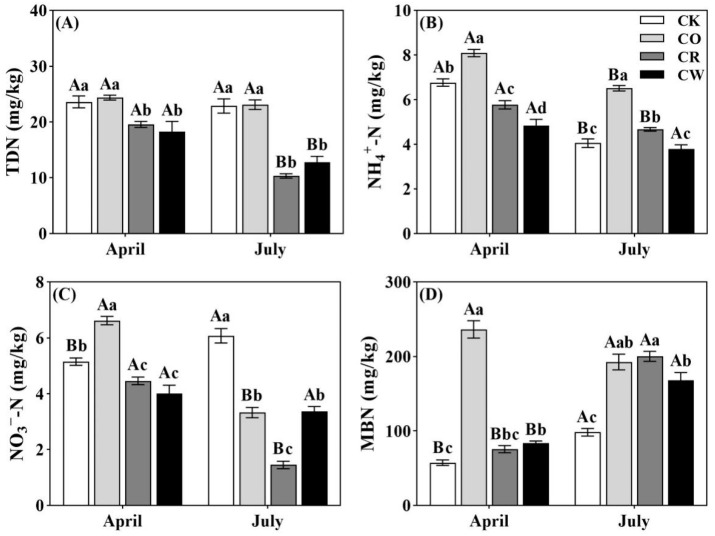
Soil N fractions content (mean ± standard deviation) of different restoration measures for arable land on purple soil slopes. CK, Control check; CO, *Camellia oleifera* monoculture; CR, *Camellia oleifera* ryegrass intercropping; CW, *Camellia oleifera* intercropping with weeds; (**A**) TDN, total soluble N; (**B**) NH_4_^+^-N, ammonium N; (**C**) NO_3_^−^-N, nitrate N; (**D**) MBN, microbial biomass N. Different lowercase letters indicate significant differences between different vegetation-restoration measures at the same time (*p* < 0.05), and different uppercase letters indicate significant differences between the same vegetation-restoration measures at different times (*p* < 0.05).

**Figure 2 plants-12-04188-f002:**
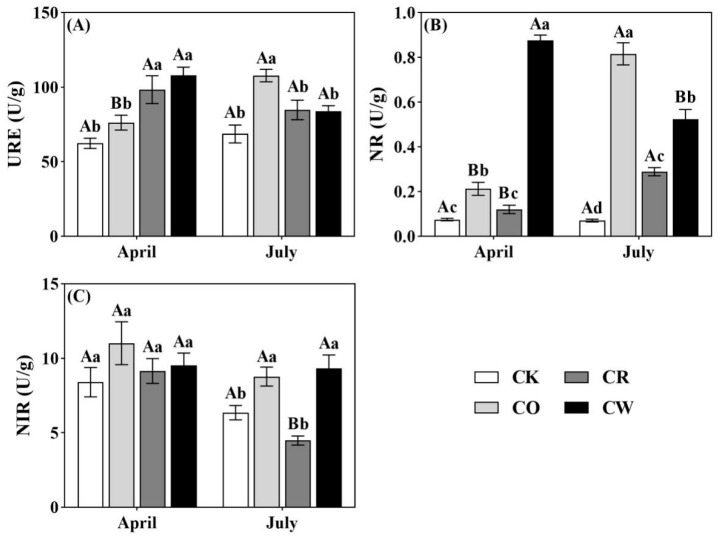
Soil enzyme activities (mean ± standard deviation) of different vegetation-restoration measures in arable land on purple soil slopes. CK, control check; CO, *Camellia oleifera* monoculture; CR, *Camellia oleifera* ryegrass intercropping; CW, *Camellia oleifera* intercropping with weeds; (**A**) URE, urease; (**B**) NR, nitrate reductase; (**C**) NIR, nitrite reductase. Different lowercase letters indicate significant (*p* < 0.05) differences between different vegetation-restoration measures at the same time, and different uppercase letters indicate significant (*p* < 0.05) differences between the same vegetation-restoration measures at different times.

**Figure 3 plants-12-04188-f003:**
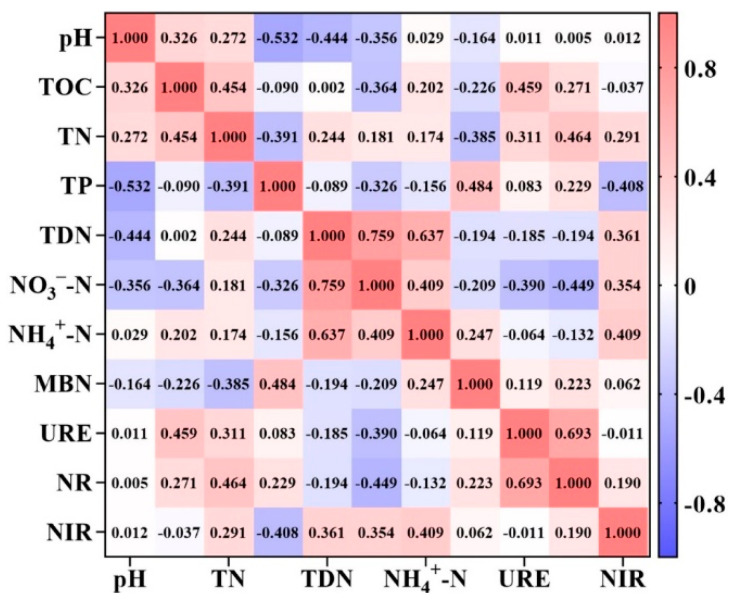
Correlation between soil enzyme activity and soil effective carbon fraction content and soil chemical properties in arable land with purple soil slopes. pH; TOC, soil total organic carbon; TN, total N; TP, total phosphorus; TDN, total soluble N; NO_3_^−^-N, nitrate N; NH_4_^+^-N, ammonium N; MBN, microbial biomass N; URE, urease; NR, nitrate reductase; NIR, nitrite reductase.

**Figure 4 plants-12-04188-f004:**
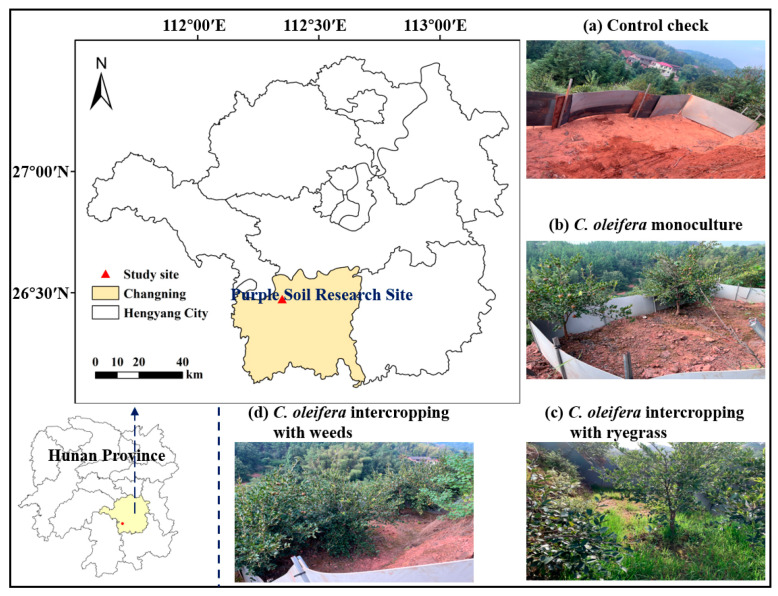
Location map of the study area. Four experimental treatments: (**a**) CK, Control check; (**b**) CO, *Camellia oleifera* monoculture; (**c**) CR, *Camellia oleifera* ryegrass intercropping; (**d**) CW, *Camellia oleifera* intercropping with weeds.

**Table 1 plants-12-04188-t001:** Vegetation restoration types and basic chemical properties of arable land on purple soil slopes in different periods.

Period	Symbol of Plot	pH	Total Soil Organic Carbon Content (g·kg^−1^)	Total Nitrogen Content (g·kg^−1^)	Total Phosphorus Content (g·kg^−1^)
Lush period (April)	CK	4.48 ± 0.020 Aa	3.20 ± 0.128 Aab	0.32 ± 0.008 Ab	0.046 ± 0.001 Bb
	CO	4.36 ± 0.038 Ab	2.82 ± 0.134 Bb	0.29 ± 0.006 Ac	0.052 ± 0.003 Ba
	CR	4.41 ± 0.008 Aab	3.36 ± 0.053 Aab	0.27 ± 0.009 Ac	0.035 ± 0.001 Bc
	CW	4.45 ± 0.004 Aa	3,63 ± 0.124 Aa	0.40 ± 0.011 Aa	0.046 ± 0.002 Bab
Wilting period (July)	CK	4.20 ± 0.027 Bb	2.56 ± 0.081 Bb	0.26 ± 0.008 Bab	0.076 ± 0.003 Ab
	CO	4.26 ± 0.013 Bb	3.26 ± 0.100 Aa	0.29 ± 0.014 Aa	0.087 ± 0.005 Aa
	CR	4.45 ± 0.019 Aa	3.41 ± 0.170 Aa	0.25 ± 0.007 Ab	0.081 ± 0.003 Aab
	CW	4.42 ± 0.026 Aa	2.44 ± 0.030 Bb	0.26 ± 0.007 Bb	0.053 ± 0.002 Ac

(mean ± standard error). CK, Control check; CO, *Camellia oleifera* monoculture; CR, *Camellia oleifera* ryegrass intercropping; CW, *Camellia oleifera* intercropping with weeds. Different lowercase letters indicate significant (*p* < 0.05) differences between different vegetation-restoration measures at the same time, and different uppercase letters indicate significant (*p* < 0.05) differences between the same vegetation-restoration measures at different times.

## Data Availability

The data presented in this study are available on request from the corresponding author. The data are not publicly available due to the funded projects not having been completed.

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
