# Peer review of "Effects of Vegetation Restoration on Soil Nitrogen Fractions and Enzyme Activities in Arable Land on Purple Soil Slopes"

_plants, 2023, doi:10.3390/plants12244188_

Round 1

Reviewer 1 Report

Comments and Suggestions for Authors

Comments: The study investigated the effects of vegetation restoration on soil nitrogen fractions and the related enzyme activities involved in the nitrogen cycle in arable land on purple soil slopes. It indicated that the cultivation of land on slopes of purple soil can affect soil nutrient content and the composition of effective nitrogen through vegetation restoration. The manuscript is well-written but suffers from a number of detail problems, which is suitable for publication after revision.

1. The presentation of the results is too lengthy, and it is suggested that a comprehensive statement is not necessary to show the change law of the main indicators.

2. I suggest that the several figures of nitrogen components can be combined into a single plot.

3. Table 1, There is a lack of pairwise comparative analysis of carbon, nitrogen and phosphorus contents among different vegetation restoration types.

4. I suggest that the results of nitrogen fractions contents should be removed from the three headings. Such as 2.1.1~2.1.4

5. There is a little grammatical error in the manuscript, please check carefully. In addition, please note whether the references match the corresponding content of the study.

6. The results are almost all general statements of the law of indicator change, and few important data are presented. It is recommended to display some important data in the text.

Comments on the Quality of English Language

The overall language of the paper is fine. Please check the format of the references and figures.

Reviewer 2 Report

Comments and Suggestions for Authors

The authors has provided an interesting and complete study which disentangle the effects of vegetation restoration on soil nitrogen fractions and the related enzyme activities involved in the nitrogen cycle in arable land on purple soil slopes. The study examines minuciosly changes in soil nutrient content, the levels of effective nitrogen fractions (i.e., microbial biomass nitrogen, ammonium nitrogen, and nitrate nitrogen), and enzyme activities involved in nitrogen mineralisation and immobilisation. The methodology used to quantify levels of total soluble nitrogen, ammonium nitrogen, and nitrate nitrogen, as well as the activities of three soil enzymes (i.e., urease, nitrate reductase, and nitrite reductase) were appropiately, providing useful information to link with the considered vegetation-restoration measures in the study (i.e., monocropping planting of tea oil, intercropping planting of tea oil and rye grass, and intercropping planting of tea oil and weeds, including tea oil-weed intercropping). The obtained results are relevant and demonstrated the effect of the vegetation restoration in the soil features. Finally, the discussion is well-structured and the used references are suitable to compare the obtained results.

However, some changes should be done before the publication:

Table 1. The word "period" should be in cappitol letters like the remaing terms in the table.

Figure 3. The caption of this figure should be in the same page of this figure.

Page 1, line 45. The subtitle "Introduction" should be at the beginning of the next page.

The authors of the species should be included the first time they appears in the text. In addition, it aldo should be included in the captions of tables and figures.

The hypothesis of the study should be specified at the end of the introduction.

Line 163 should be in the next page.

"Prism 9" and "SPSS 19.0" should be referenced in the manuscript.

References format should be checked again (e.g., reference 24 needs spaces between year, volumen and pages).

Reviewer 3 Report

Comments and Suggestions for Authors

Authors have studied different soil nitrogen fractions and soil enzyme activities in context of vegetation restoration of arable land (purple soils). The topic is sound. The manuscript is well written. All sections are in detail, everything is clear and connecting, easy to follow. Still, manuscript should be improved a little bit. I gave some comments in the Annotated manuscript (pdf).
